# Feedback Detection for Live Predictors

**Stefan Wager, Nick Chamandy, Omkar Muralidharan, and Amir Najmi**
swager@stanford.edu, {chamandy, omuralidharan, amir}@google.com
*Stanford University and Google, Inc.*

## Abstract

A predictor that is deployed in a live production system may perturb the features it uses to make predictions. Such a feedback loop can occur, for example, when a model that predicts a certain type of behavior ends up causing the behavior it predicts, thus creating a self-fulfilling prophecy. In this paper we analyze predictor feedback detection as a causal inference problem, and introduce a local randomization scheme that can be used to detect non-linear feedback in real-world problems. We conduct a pilot study for our proposed methodology using a predictive system currently deployed as a part of a search engine.

## 1   Introduction

When statistical predictors are deployed in a live production environment, feedback loops can become a concern. Predictive models are usually tuned using training data that has not been influenced by the predictor itself; thus, most real-world predictors cannot account for the effect they themselves have on their environment. Consider the following caricatured example: A search engine wants to train a simple classifier that predicts whether a search result is "newsy" or not, meaning that the search result is relevant for people who want to read the news. This classifier is trained on historical data, and learns that high click-through rate (CTR) has a positive association with "newsiness." Problems may arise if the search engine deploys the classifier, and starts featuring search results that are predicted to be newsy for some queries: promoting the search result may lead to a higher CTR, which in turn leads to higher newsiness predictions, which makes the result be featured even more.

If we knew beforehand all the channels through which predictor feedback can occur, then detecting feedback would not be too difficult. For example, in the context of the above example, if we knew that feedback could *only* occur through some changes to the search result page that were directly triggered by our model, then we could estimate feedback by running small experiments where we turn off these triggering rules. However, in large industrial systems where networks of classifiers all feed into each other, we can no longer hope to understand *a priori* all the ways in which feedback may occur. We need a method that lets us detect feedback from sources we might not have even known to exist.

This paper proposes a simple method for detecting feedback loops from unknown sources in live systems. Our method relies on artificially inserting a small amount of noise into the predictions made by a model, and then measuring the effect of this noise on future predictions made by the model. If future model predictions change when we add artificial noise, then our system has feedback.

To understand how random noise can enable us to detect feedback, suppose that we have a model with predictions $\hat{y}$ in which tomorrow's prediction $\hat{y}^{(t+1)}$ has a linear feedback dependence on today's prediction $\hat{y}^{(t)}$: if we increase $\hat{y}^{(t)}$ by $\delta$, then $\hat{y}^{(t+1)}$ increases by $\beta\,\delta$ for some $\beta \in \mathbb{R}$. Intuitively, we should be able to fit this slope $\beta$ by perturbing $\hat{y}^{(t)}$ with a small amount of noise $\nu \sim \mathcal{N}\left(0,\,\sigma_\nu^2\right)$ and then regressing the new $\hat{y}^{(t+1)}$ against the noise; the reason least squares should work here is that the noise $\nu$ is independent of all other variables by construction. The main contribution of this paper is to turn this simple estimation idea into a general procedure that can be used to detect feedback in realistic problems where the feedback has non-linearities and jumps.

**Counterfactuals and Causal Inference**   Feedback detection is a problem in causal inference. A model suffers from feedback if the predictions it makes today affect the predictions it will make tomorrow. We are thus interested in discovering a causal relationship between today's and tomorrow's predictions; simply detecting a correlation is not enough. The distinction between causal and associational inference is acute in the case of feedback: today's and tomorrow's predictions are almost always strongly correlated, but this correlation by no means implies any causal relationship.

In order to discover causal relationships between consecutive predictions, we need to use some form of randomized experimentation. In our case, we add a small amount of random noise to our predictions. Because the noise is fully artificial, we can reasonably ask counterfactual questions of the type: "How would tomorrow's predictions have changed if we added more/less noise to the predictions today?" The noise acts as an independent instrument that lets us detect feedback. We frame our analysis in terms of a potential outcomes model that asks how the world would have changed had we altered a treatment; in our case, the treatment is the random noise we add to our predictions. This formalism, often called the Rubin causal model [1], is regularly used for understanding causal inference [2, 3, 4]. Causal models are useful for studying the behavior of live predictive systems on the internet, as shown by, e.g., the recent work of Bottou et al. [5] and Chan et al. [6].

**Outline and Contributions**   In order to define a rigorous feedback detection procedure, we need to have a precise notion of what we mean by feedback. Our first contribution is thus to provide such a model by defining statistical feedback in terms of a potential outcomes model (Section 2). Given this feedback model, we propose a local noising scheme that can be used to fit feedback functions with non-linearities and jumps (Section 4). Before presenting general version of our approach, however, we begin by discussing the linear case in Section 3 to elucidate the mathematics of feedback detection: as we will show, the problem of linear feedback detection using local perturbations reduces to linear regression. Finally, in Section 5 we conduct a pilot study based on a predictive model currently deployed as a part of a search engine.

## 2   Feedback Detection for Statistical Predictors

Suppose that we have a model that makes predictions $\hat{y}_i^{(t)}$ in time periods $t = 1, 2, \dots$ for examples $i = 1, \dots, n$. The predictive model itself is taken as given; our goal is to understand feedback effects between consecutive pairs of predictions $\hat{y}_i^{(t)}$ and $\hat{y}_i^{(t+1)}$. We define statistical feedback in terms of counterfactual reasoning: we want to know what would have happened to $\hat{y}_i^{(t+1)}$ had $\hat{y}_i^{(t)}$ been different. We use potential outcomes notation [e.g., 7] to distinguish between counterfactuals: let $\hat{y}_i^{(t+1)}[y_i^{(t)}]$ be the predictions our model *would have made* at time $t + 1$ if we had published $y_i^{(t)}$ as our time-$t$ prediction. In practice we only get to observe $\hat{y}_i^{(t+1)}[y_i^{(t)}]$ for a single $y_i^{(t)}$; all other values of $\hat{y}_i^{(t+1)}[y_i^{(t)}]$ are counterfactual. We also consider $\hat{y}_i^{(t+1)}[\varnothing]$, the prediction our model would have made at time $t + 1$ if the model never made any of its predictions public and so did not have the chance to affect its environment. With this notation, we define feedback as

$$\text{feedback}_i^{(t)} = \hat{y}_i^{(t+1)}[\hat{y}_i^{(t)}] - \hat{y}_i^{(t+1)}[\varnothing], \tag{1}$$

i.e., the difference between the predictions our model actually made and the predictions it would have made had it not had the chance to affect its environment by broadcasting predictions in the past. Thus, statistical feedback is a difference in potential outcomes.

**An additive feedback model** In order to get a handle on feedback as defined above, we assume that feedback enters the model additively: $\hat{y}_i^{(t+1)}[y_i^{(t)}] = \hat{y}_i^{(t+1)}[\varnothing] + f(y_i^{(t)})$, where $f$ is a feedback function, and $y_i^{(t)}$ is the prediction published at time $t$. In other words, we assume that the predictions made by our model at time $t+1$ are the sum of the prediction the model would have made if there were no feedback, plus a feedback term that only depends on the previous prediction made by the model. Our goal is to estimate the feedback function $f$.

**Artificial noising for feedback detection** The relationship between $\hat{y}_i^{(t)}$ and $\hat{y}_i^{(t+1)}$ can be influenced by many things, such as trends, mean reversion, random fluctuations, as well as feedback. In order to isolate the effect of feedback, we need to add some noise to the system to create a situation that resembles a randomized experiment. Ideally, we might hope to sometimes turn our predictive system off in order to get estimates of $\hat{y}_i^{(t)}[\varnothing]$. However, predictive models are often deeply integrated into large software systems, and it may not be clear what the correct system behavior would be if we turned the predictor off. To side-step this concern, we randomize our system by adding artificial noise to predictions: at time $t$, instead of deploying the prediction $\hat{y}_i^{(t)}$, we deploy $\check{y}_i^{(t)} = \hat{y}_i^{(t)} + \nu_i^{(t)}$, where $\nu_i^{(t)} \overset{iid}{\sim} N$ is artificial noise drawn from some distribution $N$. Because the noise $\nu_i^{(t)}$ is independent from everything else, it puts us in a randomized experimental setup that allows us to detect feedback as a causal effect. If the time $t+1$ prediction $\hat{y}_i^{(t+1)}$ is affected by $\nu_i^{(t)}$, then our system must have feedback because the only way $\nu_i^{(t)}$ can influence $\hat{y}_i^{(t+1)}$ is through the interaction between our model predictions and the surrounding environment at time $t$.

**Local average treatment effect** In practice, we want the noise $\nu_i^{(t)}$ to be small enough that it does not disturb the regular operation of the predictive model too much. Thus, our experimental setup allows us to measure feedback as a local average treatment effect [4], where the artificial noise $\nu_i^{(t)}$ acts as a continuous treatment. Provided our additive model holds, we can then piece together these local treatment effects into a single global feedback function $f$.

## 3 Linear Feedback

We begin with an analysis of linear feedback problems; the linear setup allows us to convey the main insights with less technical overhead. We discuss the non-linear case in Section 4. Suppose that we have some natural process $x^{(1)}$, $x^{(2)}$, ... and a predictive model of the form $\hat{y} = w \cdot x$. (Suppose for notational convenience that $x$ includes the constant, and the intercept term is folded into $w$.) For our purposes, $w$ is fixed and known; for example, $w$ may have been set by training on historical data. At some point, we ship a system that starts broadcasting the predictions $\hat{y} = w \cdot x$, and there is a concern that the act of broadcasting the $\hat{y}$ may perturb the underlying $x^{(t)}$ process. Our goal is to detect any such feedback. Following earlier notation we write $\hat{y}_i^{(t+1)}[\hat{y}_i^{(t)}] = w \cdot x_i^{(t+1)}[\hat{y}_i^{(t)}]$ for the time $t+1$ variables perturbed by feedback, and $\hat{y}_i^{(t+1)}[\varnothing] = w \cdot x_i^{(t+1)}[\varnothing]$ for the counterparts we would have observed without any feedback.

In this setup, any effect of $\hat{y}_i^{(t)}$ on $x_i^{(t+1)}[\hat{y}_i^{(t)}]$ is feedback. A simple way to constrain this relationship is using a linear model $x_i^{(t+1)}[\hat{y}_i^{(t)}] = x_i^{(t+1)}[\varnothing] + \hat{y}_i^{(t)}\gamma$. In other words, we assume that $x_i^{(t+1)}[\hat{y}_i^{(t)}]$ is perturbed by an amount that scales linearly with $\hat{y}_i^{(t)}$. Given this simple model, we find that:

$$\hat{y}_i^{(t+1)}[\hat{y}_i^{(t)}] = \hat{y}_i^{(t+1)}[\varnothing] + w \cdot \gamma \,\hat{y}_i^{(t)}, \tag{2}$$

and so $f(y) = \beta\, y$ with $\beta = w \cdot \gamma$; $f$ is the feedback function we want to fit.

We cannot work with (2) directly, because $\hat{y}_i^{(t+1)}{}_{[\varnothing]}$ is not observed. In order to get around this problem, we add artificial noise to our predictions: at time $t$, we publish predictions $\breve{y}_i^{(t)} = \hat{y}_i^{(t)} + \nu_i^{(t)}$ instead of the raw predictions $\hat{y}_i^{(t)}$. As argued in Section 2, this method lets us detect feedback because $\hat{y}_i^{(t+1)}$ can only depend on $\nu_i^{(t)}$ through a feedback mechanism, and so any relationship between $\hat{y}_i^{(t+1)}$ and $\nu_i^{(t)}$ must be a symptom of feedback.

**A Simple Regression Approach**  With the linear feedback model (2), the effect of $\nu_i^{(t)}$ on $\hat{y}_i^{(t+1)}$ is $\hat{y}_i^{(t+1)}{}_{[\hat{y}_i^{(t)}+\nu_i^{(t)}]} = \hat{y}_i^{(t+1)}{}_{[\hat{y}_i^{(t)}]} + \beta\,\nu_i^{(t)}$. This relationship suggests that we should be able to recover $\beta$ by regressing $\hat{y}_i^{(t+1)}$ against the added noise $\nu_i^{(t)}$. The following result confirms this intuition.

**Theorem 1.** *Suppose that* (2) *holds, and that we add noise* $\nu_i^{(t)}$ *to our time $t$ predictions. If we estimate $\beta$ using linear least squares*

$$\hat{\beta} = \frac{\widehat{\text{Cov}}\left[\hat{y}_i^{(t+1)}{}_{[\hat{y}_i^{(t)}+\nu_i^{(t)}]},\, \nu_i^{(t)}\right]}{\widehat{\text{Var}}\left[\nu_i^{(t)}\right]},\ \textit{then}\ \sqrt{n}\left(\hat{\beta}-\beta\right) \Rightarrow \mathcal{N}\left(0,\ \frac{\text{Var}\left[\hat{y}_i^{(t+1)}{}_{[\hat{y}_i^{(t)}]}\right]}{\sigma_\nu^2}\right), \quad (3)$$

*where $\sigma_\nu^2 = \text{Var}\left[\nu_i^{(t)}\right]$ and $n$ is the number of examples to which we applied our predictor.*

Theorem 1 gives us a baseline understanding for the difficulty of the feedback detection problem: the precision of our feedback estimates scales as the ratio of the artificial noise $\sigma_\nu^2$ to natural noise $\text{Var}[\hat{y}_i^{(t+1)}{}_{[\hat{y}_i^{(t)}]}]$. Note that the proof of Theorem 1 assumes that we only used predictions from a single time period $t+1$ to fit feedback, and that the raw predictions $\hat{y}_i^{(t+1)}{}_{[\hat{y}_i^{(t)}]}$ are all independent. If we relax these assumptions we get a regression problem with correlated errors, and need to be more careful with technical conditions.

**Efficiency and Conditioning**  The simple regression model (3) treats the term $\hat{y}_i^{(t+1)}{}_{[\hat{y}_i^{(t)}]}$ as noise. This is quite wasteful: if we know $\hat{y}_i^{(t)}$ we usually have a fairly good idea of what $\hat{y}_i^{(t+1)}{}_{[\hat{y}_i^{(t)}]}$ should be, and not using this information needlessly inflates the noise. Suppose that we knew the function[1]

$$\mu(y) := \mathbb{E}\left[\hat{y}_i^{(t+1)}{}_{[\hat{y}_i^{(t)}]}\ \middle|\ \hat{y}_i^{(t)} = y\right]. \tag{4}$$

Then, we could write our feedback model as

$$\hat{y}_i^{(t+1)}{}_{[\hat{y}_i^{(t)}+\nu_i^{(t)}]} = \mu\left(\hat{y}_i^{(t)}\right) + \left(\hat{y}_i^{(t+1)}{}_{[\hat{y}_i^{(t)}]} - \mu\left(\hat{y}_i^{(t)}\right)\right) + \beta\,\nu_i^{(t)}, \tag{5}$$

where $\mu(\hat{y}_i^{(t)})$ is a known offset. Extracting this offset improves the precision of our estimate for $\hat{\beta}$.

**Theorem 2.** *Under the conditions of Theorem 1 suppose that the function $\mu$ from (4) is known and that the $\hat{y}_i^{(t+1)}$ are all independent of each other conditional on $\hat{y}_i^{(t)}$. Then, given the information available at time $t$, the estimate*

$$\hat{\beta}^* = \frac{\widehat{\text{Cov}}\left[\hat{y}_i^{(t+1)}{}_{[\hat{y}_i^{(t)}+\nu_i^{(t)}]} - \mu\left(\hat{y}_i^{(t)}\right),\, \nu_i^{(t)}\right]}{\widehat{\text{Var}}\left[\nu_i^{(t)}\right]}\quad \textit{has asymptotic distribution} \tag{6}$$

$$\sqrt{n}\left(\hat{\beta}^*-\beta\right) \Rightarrow \mathcal{N}\left(0,\ \frac{\mathbb{E}\left[\text{Var}\left[\hat{y}_i^{(t+1)}{}_{[\hat{y}_i^{(t)}]}\ \middle|\ \hat{y}_i^{(t)}\right]\right]}{\sigma_\nu^2}\right). \tag{7}$$

*Moreover, if the variance of $\eta_i^{(t)} = \hat{y}_i^{(t+1)}[\hat{y}_i^{(t)}] - \mu(\hat{y}_i^{(t)})$ does not depend on $\hat{y}_i^{(t)}$, then $\hat{\beta}^*$ is the best linear unbiased estimator of $\beta$.*

Theorem 2 extends the general result from above that the precision with which we can estimate feedback scales as the ratio of artificial noise to natural noise. The reason why $\hat{\beta}^*$ is more efficient than $\hat{\beta}$ is that we managed to condition away some of the natural noise, and reduced the variance of our estimate for $\beta$ by

$$\text{Var}\left[\mu\left(\hat{y}_i^{(t)}\right)\right] = \text{Var}\left[\hat{y}_i^{(t+1)}[\hat{y}_i^{(t)}]\right] - \mathbb{E}\left[\text{Var}\left[\hat{y}_i^{(t+1)}[\hat{y}_i^{(t)}] \mid \hat{y}_i^{(t)}\right]\right]. \tag{8}$$

In other words, the variance reduction we get from $\hat{\beta}^*$ directly matches the amount of variability we can explain away by conditioning. The estimator (6) is not practical as stated, because it requires knowledge of the unknown function $\mu$ and is restricted to the case of linear feedback. In the next section, we generalize this estimator into one that does not require prior knowledge of $\mu$ and can handle non-linear feedback.

## 4    Fitting Non-Linear Feedback

Suppose now that we have the same setup as in the previous section, except that now feedback has a non-linear dependence on the prediction: $\hat{y}_i^{(t+1)}[\hat{y}_i^{(t)}] = \hat{y}_i^{(t+1)}[\varnothing] + f(\hat{y}_i^{(t)})$ for some arbitrary function $f$. For example, in the case of a linear predictive model $\hat{y} = w \cdot x$, this kind of feedback could arise if we have feature feedback $x_i^{(t+1)}[\hat{y}_i^{(t)}] = x_i^{(t+1)}[\varnothing] + f_{(x)}(\hat{y}_i^{(t)})$; the feedback function then becomes $f(\cdot) = w \cdot f_{(x)}(\cdot)$. When we add noise $\nu_i^{(t)}$ to the above predictions, we only affect the feedback term $f(\cdot)$:

$$\hat{y}_i^{(t+1)}[\hat{y}_i^{(t)}+\nu_i^{(t)}] - \hat{y}_i^{(t+1)}[\hat{y}_i^{(t)}] = f\left(\hat{y}_i^{(t)} + \nu_i^{(t)}\right) - f\left(\hat{y}_i^{(t)}\right). \tag{9}$$

Thus, by adding artificial noise $\nu_i^{(t)}$, we are able to cancel out the nuisance terms, and isolate the feedback function $f$ that we want to estimate. We cannot use (9) in practice, though, as we can only observe one of $\hat{y}_i^{(t+1)}[\hat{y}_i^{(t)}+\nu_i^{(t)}]$ or $\hat{y}_i^{(t+1)}[\hat{y}_i^{(t)}]$ in reality; the other one is counterfactual. We can get around this problem by conditioning on $\hat{y}_i^{(t)}$ as in Section 3. Let

$$\mu\left(y\right) = \mathbb{E}\left[\hat{y}_i^{(t+1)}[\hat{y}_i^{(t)}+\nu_i^{(t)}] \mid \hat{y}_i^{(t)} = y\right] \tag{10}$$

$$= t\left(y\right) + \varphi_N * f\left(y\right), \text{ where } t\left(y\right) = \mathbb{E}\left[\hat{y}_i^{(t+1)}[\varnothing] \mid \hat{y}_i^{(t)} = y\right]$$

is a term that captures trend effects that are not due to feedback. The $*$ denotes convolution:

$$\varphi_N * f\left(y\right) = \mathbb{E}\left[f\left(\hat{y}_i^{(t)} + \nu_i^{(t)}\right) \mid \hat{y}_i^{(t)} = y\right] \text{ with } \nu_i^{(t)} \sim N. \tag{11}$$

Using the conditional mean function $\mu$ we can write our expression of interest as

$$\hat{y}_i^{(t+1)}[\hat{y}_i^{(t)}+\nu_i^{(t)}] - \mu\left(\hat{y}_i^{(t)}\right) = f\left(\hat{y}_i^{(t)} + \nu_i^{(t)}\right) - \varphi_N * f\left(\hat{y}_i^{(t)}\right) + \eta_i^{(t)}, \tag{12}$$

where $\eta_i^{(t)} := \hat{y}_i^{(t+1)}[\varnothing] - t\left(\hat{y}_i^{(t)}\right)$. If we have a good idea of what $\mu$ is, the left-hand side can be measured, as it only depends on $\hat{y}_i^{(t+1)}[\hat{y}_i^{(t)}+\nu_i^{(t)}]$ and $\hat{y}_i^{(t)}$. Meanwhile, conditional on $\hat{y}_i^{(t)}$, the first two terms on the right-hand side only depend on $\nu_i^{(t)}$, while $\eta_i^{(t)}$ is independent of $\nu_i^{(t)}$ and mean-zero. The upshot is that we can treat (12) as a regression problem where $\eta_i^{(t)}$ is noise. In practice, we estimate $\mu$ from an auxiliary problem where we regress $\hat{y}_i^{(t+1)}[\hat{y}_i^{(t)}+\nu_i^{(t)}]$ against $\hat{y}_i^{(t)}$.

**A Pragmatic Approach** There are many possible approaches to solving the non-parametric system of equations (12) for $f$ [e.g., 8, Chapter 5]. Here, we take a pragmatic approach, and constrain ourselves to solutions of the form $\hat{\mu}(y) = \hat{\beta}_\mu \cdot b_\mu(y)$ and $\hat{f}(y) = \hat{\beta}_f \cdot b_f(y)$, where $b_\mu : \mathbb{R} \to \mathbb{R}^{p_\mu}$ and $b_f : \mathbb{R} \to \mathbb{R}^{p_f}$ are predetermined basis expansions. This approach transforms our problem into an ordinary least-squares problem, and works well in terms of producing reasonable feedback estimates in real-world problems (see Section 5). If this relation in fact holds for some values $\beta_\mu$ and $\beta_f$, the result below shows that we can recover $\beta_f$ by least-squares.

**Theorem 3.** *Suppose that $\beta_\mu$ and $\beta_f$ are defined as above, and that we have an unbiased estimator $\hat{\beta}_\mu$ of $\beta_\mu$ with variance $V_\mu = \mathrm{Var}[\hat{\beta}_\mu]$. Then, if we fit $\beta_f$ by least squares using (12) as described in Appendix A, the resulting estimate $\hat{\beta}_f$ is unbiased and has variance*

$$\mathrm{Var}\left[\hat{\beta}_f\right] = \left(X_f^\intercal X_f\right)^{-1} X_f^\intercal \left(V_Y + X_\mu V_\mu X_\mu^\intercal\right) X_f \left(X_f^\intercal X_f\right)^{-1}, \tag{13}$$

*where the design matrices $X_\mu$ and $X_f$ are defined as*

$$X_\mu = \begin{pmatrix} \vdots \\ b_\mu^\intercal\left(\hat{y}_i^{(t)}\right) \\ \vdots \end{pmatrix} \text{ and } X_f = \begin{pmatrix} \vdots \\ b_f^\intercal\left(\hat{y}_i^{(t)} + \nu_i^{(t)}\right) - (\varphi_N * b_f)^\intercal\left(\hat{y}_i^{(t)}\right) \\ \vdots \end{pmatrix} \tag{14}$$

*and $V_Y$ is a diagonal matrix with $(V_Y)_{ii} = \mathrm{Var}\left[\hat{y}_i^{(t+1)}{}_{[\hat{y}_i^{(t)}]} \mid \hat{y}_i^{(t)}\right]$.*

In the case where our spline model is misspecified, we can obtain a similar result using methods due to Huber [9] and White [10]. In practice, we can treat $\hat{\beta}_\mu$ as known since fitting $\mu(\cdot)$ is usually easier than fitting $f(\cdot)$: estimating $\mu(\cdot)$ is just a smoothing problem whereas estimating $f(\cdot)$ requires fitting differences. If we also treat the errors $\eta_i^{(t)}$ in (12) as roughly homoscedatic, (13) reduces to

$$\mathrm{Var}\left[\hat{\beta}_f\right] \approx \frac{\mathbb{E}\left[\mathrm{Var}\left[\hat{y}_i^{(t+1)}{}_{[\hat{y}_i^{(t)}]} \mid \hat{y}_i^{(t)}\right]\right]}{n\,\mathbb{E}\left[\|s_i\|_2^2\right]}, \text{ where } s_i = b_f\left(\hat{y}_i^{(t)} + \nu_i^{(t)}\right) - \varphi_N * b_f\left(\hat{y}_i^{(t)}\right). \tag{15}$$

This simplified form again shows that the precision of our estimate of $f(\cdot)$ scales roughly as the ratio of the variance of the artificial noise $\nu_i^{(t)}$ to the variance of the natural noise.

**Our Method in Practice** For convenience, we summarize the steps needed to implement our method here: **(1)** At time $t$, compute model predictions $\hat{y}_i^{(t)}$ and draw noise terms $\nu_i^{(t)} \overset{iid}{\sim} N$ for some noise distribution $N$. Deploy predictions $\breve{y}_i^{(t)} = \hat{y}_i^{(t)} + \nu_i^{(t)}$ in the live system. **(2)** Fit a non-parametric least-squares regression of $\hat{y}_i^{(t+1)}{}_{[\hat{y}_i^{(t)}+\nu_i^{(t)}]} \sim \mu\left(\hat{y}_i^{(t)}\right)$ to learn the function $\mu(y) := \mathbb{E}\left[\hat{y}_i^{(t+1)}{}_{[\hat{y}_i^{(t)}+\nu_i^{(t)}]} \mid \hat{y}_i^{(t)} = y\right]$. We use the R formula notation, where $a \sim g(b)$ means that we want to learn a function $g(b)$ that predicts $a$. **(3)** Set up the non-parametric least-squares regression problem

$$\hat{y}_i^{(t+1)}{}_{[\hat{y}_i^{(t)}+\nu_i^{(t)}]} - \mu\left(\hat{y}_i^{(t)}\right) \sim f\left(\hat{y}_i^{(t)} + \nu_i^{(t)}\right) - \varphi_N * f\left(\hat{y}_i^{(t)}\right), \tag{16}$$

where the goal is to learn $f$. Here, $\varphi_N$ is the density of $\nu_i^{(t)}$, and $*$ denotes convolution. In Appendix A we show how to carry out these steps using standard R libraries.

The resulting function $f(y)$ is our estimate of feedback: If we make a prediction $\breve{y}_i^{(t)}$ at time $t$, then our time $t+1$ prediction will be boosted by $f(\breve{y}_i^{(t)})$. The above equation only depends on $\hat{y}_i^{(t)}$, $\nu_i^{(t)}$,

and $\hat{y}_i^{(t+1)}{}_{[\hat{y}_i^{(t)}+\nu_i^{(t)}]}$, which are all quantities that can be observed in the context of an experiment with noised predictions. Note that as we only fit $f$ using the differences in (16), the intercept of $f$ is not identifiable. We fix the intercept (rather arbitrarily) by setting the average fitted feedback over all training examples to 0; we do not include an intercept term in the basis $b_f$.

**Choice of Noising Distribution**  Adding noise to deployed predictions often has a cost that may depend on the shape of the noise distribution $N$. A good choice of $N$ should reflect this cost. For example, if the practical cost of adding noise only depends on the largest amount of noise we ever add, then it may be a good idea to draw $\nu_i^{(t)}$ uniformly at random from $\{\pm\varepsilon\}$ for some $\varepsilon > 0$. In our experiments, we draw noise from a Gaussian distribution $\nu_i^{(t)} \sim \mathcal{N}(0, \sigma_\nu^2)$.

# 5  A Pilot Study

The original motivation for this research was to develop a methodology for detecting feedback in real-world systems. Here, we present results from a pilot study, where we added signal to historical data that we believe should emulate actual feedback. The reason for monitoring feedback on this system is that our system was about to be more closely integrated with other predictive systems, and there was a concern that the integration could induce bad feedback loops. Having a reliable method for detecting feedback would provide us with an early warning system during the integration.

The predictive model in question is a logistic regression classifier. We added feedback to historical data collected from log files according to half a dozen rules of the form "if $a_i^{(t)}$ is high and $\check{y}_i^{(t)} > 0$, then increase $a_i^{(t+1)}$ by a random amount"; here $\check{y}_i^{(t)}$ is the time-$t$ prediction deployed by our system (in log-odds space) and $a_i^{(t)}$ is some feature with a positive coefficient. These feedback generation rules do not obey the additive assumption. Thus our model is misspecified in the sense that there is no function $f$ such that a current prediction $\check{y}_i^{(t)}$ increased the log-odds of the next prediction by $f(\check{y}_i^{(t)})$, and so this example can be taken as a stretch case for our method.

Our dataset had on the order of 100,000 data points, half of which were used for fitting the model itself and half of which were used for feedback simulation. We generated data for 5 simulated time periods, adding noise with $\sigma_\nu = 0.1$ at each step, and fit feedback using a spline basis discussed in Appendix B. The "true feedback" curve was obtained by fitting a spline regression to the additive feedback model by looking at the unobservable $\hat{y}_i^{(t+1)}{}_{[\varnothing]}$; we used a $df = 5$ natural spline with knots evenly spread out on $[-9, 3]$ in log-odds space plus a jump at 0.

For our classifier of interest, we have fairly strong reasons to believe that the feedback function may have a jump at zero, but probably shouldn't have any other big jumps. Assuming that we know *a priori* where to look for jumps does not seem to be too big a problem for the practical applications we have considered. Results for feedback detection are shown in Figure 1. Although the fit is not perfect, we appear to have successfully detected the shape of feedback. The error bars for estimated feedback were obtained using a non-parametric bootstrap [11] for which we resampled pairs of (current, next) predictions.

This simulation suggests that our method can be used to accurately detect feedback on scales that may affect real-world systems. Knowing that we can detect feedback is reassuring from an engineering point of view. On a practical level, the feedback curve shown in Figure 1 may not be too big a concern *yet*: the average feedback is well within the noise level of the classifier. But in large-scale systems the ways in which a model interacts with its environment is always changing, and it is entirely plausible that some innocuous-looking change in the future would increase the amount of feedback. Our methodology provides us with a way to continuously monitor how feedback is affected by changes to the system, and can alert us to changes that cause problems. In Appendix B, we show some simulations with a wider range of effect sizes.

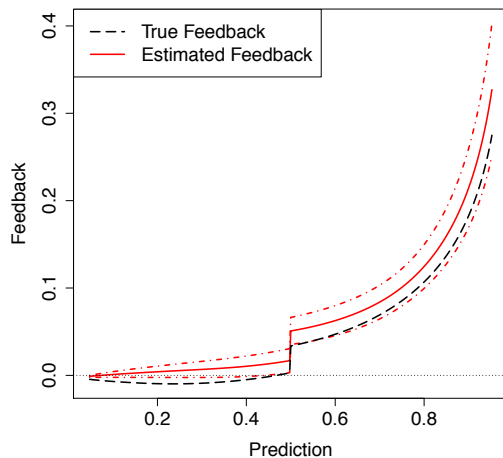

Figure 1: Simulation aiming to replicate realistic feedback in a real-world classifier. The red solid line is our feedback estimate; the black dashed line is the best additive approximation to the true feedback. The $x$-axis shows predictions in probability space; the $y$ axis shows feedback in log-odds space. The error bars indicate pointwise confidence intervals obtained using a non-parametric bootstrap with $B = 10$ replicates, and stretch 1 SE in each direction. Further experiments are provided in Appendix B.

## 6  Conclusion

In this paper, we proposed a randomization scheme that can be used to detect feedback in real-world predictive systems. Our method involves adding noise to the predictions made by the system; this noise puts us in a randomized experimental setup that lets us measure feedback as a causal effect. In general, the scale of the artificial noise required to detect feedback is smaller than the scale of the natural predictor noise; thus, we can deploy our feedback detection method without disturbing our system of interest too much. The method does not require us to make hypotheses about the mechanism through which feedback may propagate, and so it can be used to continuously monitor predictive systems and alert us if any changes to the system lead to an increase in feedback.

**Related Work**   The interaction between models and the systems they attempt to describe has been extensively studied across many fields. Models can have different kinds of feedback effects on their environments. At one extreme of the spectrum, models can become self-fulfilling prophecies: for example, models that predict economic growth may in fact cause economic growth by instilling market confidence [12, 13]. At the other end, models may distort the phenomena they seek to describe and therefore become invalid. A classical example of this is a concern that any metric used to regulate financial risk may become invalid as soon as it is widely used, because actors in the financial market may attempt to game the metric to avoid regulation [14]. However, much of the work on model feedback in fields like finance, education, or macro-economic theory has focused on negative results: there is an emphasis on understanding when feedback can happen and promoting awareness about how feedback can interact with policy decisions, but there does not appear to be much focus on actually fitting feedback. One notable exception is a paper by Akaike [15], who showed how to fit cross-component feedback in a system with many components; however, he did not add artificial noise to the system, and so was unable to detect feedback of a single component on itself.

**Acknowledgments**   The authors are grateful to Alex Blocker, Randall Lewis, and Brad Efron for helpful suggestions and interesting conversations. S. W. is supported by a B. C. and E. J. Eaves Stanford Graduate Fellowship.

## Footnotes

[1]In practice we do not know $\mu$, but we can estimate it; see Section 4.

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
