[Supplementary Material]

## A Fitting Non-Linear Feedback by Ordinary Least Squares Regression

Carrying out the fitting procedure outlined in Section A is straight-forward using standard R functions if we are willing to construct $\mu(\cdot)$ and $f(\cdot)$ using pre-specified basis expansions

$$\hat{\mu}(y) = \hat{\beta}_\mu \cdot b_\mu(y) \text{ and } \hat{f}(y) = \hat{\beta}_f \cdot b_f(y). \tag{17}$$

Recall that $b_f$ cannot have an intercept, as it would not be identifiable. We first need to construct the design matrices

$$X_\mu = \begin{pmatrix} \vdots \\ b_\mu^\intercal \left( \hat{y}_i^{(t)} \right) \\ \vdots \end{pmatrix} \text{ and } X_f = \begin{pmatrix} \vdots \\ b_f^\intercal \left( \hat{y}_i^{(t)} + \nu_i^{(t)} \right) - (\varphi_N * b_f)^\intercal \left( \hat{y}_i^{(t)} \right) \\ \vdots \end{pmatrix} \tag{18}$$

from (14). Constructing $X_\mu$ just involves choosing a basis function; however, evaluating

$$\gamma_i = (\varphi_N * b_f)^\intercal \left( \hat{y}_i^{(t)} \right) \tag{19}$$

for each row of $X_f$ can be computationally intensive if we are not careful. In particular, evaluating $\gamma_i$ by numerical integration separately for each $i$ can be painfully slow. A more efficient way to compute $\gamma_i$ is to evaluate $(\varphi_N * b_f)(y)$ over a grid of $y$-values in a single pass using the fast Fourier transform (e.g., by using `convolve` in R), and then to linearly interpolate the result onto the real line (e.g., using `approxfun`).

Once we have computed these design matrices, we can estimate $\hat{\beta}_\mu$ and $\hat{\beta}_f$ by solving the linear regression problems

$$Y \sim X_\mu \hat{\beta}_\mu \tag{20}$$

and

$$\left( Y - X_\mu \hat{\beta}_\mu \right) \sim X_f \hat{\beta}_f, \tag{21}$$

where $Y$ is just a vector with entries $\hat{y}_i^{(t+1)}{}_{[\hat{y}_i^{(t)} + \nu_i^{(t)}]}$. Notice that this whole procedure only requires knowledge of $X_\mu$, $X_f$, and the noised new predictions $\hat{y}_i^{(t+1)}{}_{[\hat{y}_i^{(t)} + \nu_i^{(t)}]}$; we never reference the counterfactual predictions $\hat{y}_i^{(t+1)}{}_{[\hat{y}_i^{(t)}]}$ or unobservable predictions $\hat{y}_i^{(t+1)}{}_{[\varnothing]}$.

In practice, most of the errors in our procedure come from the difference equation (21) and not from the conditional mean regression (20). Thus, when our model is well-specified and the additivity assumption holds, we can get good estimates for the accuracy of $f$ by looking at the parametric standard error estimates provided by `lm` from fitting (21); this is what we did for the simulations presented in Figure 2. In case of model misspecification, however, parametric confidence intervals can break down and it is better to use non-parametric methods such as the bootstrap. We used a non-parametric bootstrap for the logs simulation presented in Section 5.

## B Simulation Experiments

Here, we present a collection of simulation experiments, the results of which are given in Figure 2. These examples are all logistic regression examples with additive feedback in log-odds space. In the plots, the $y$-axis shows feedback in log-odds space, whereas the $x$-axis shows deployed predictions in probability space.

The simulations all had $n = 100,000$ (old prediction, new prediction) pairs. The predictions had natural noise with standard error $\sigma = 0.5$, i.e., the $i^{th}$ pair was centered at $\mu_i$ and distributed as $\hat{y}_i^{(t)}, \hat{y}_i^{(t+1)}{}_{[\varnothing]} \overset{iid}{\sim} \mathcal{N}\left( \mu_i, 0.5^2 \right)$. We added Gaussian noise with $\sigma_\nu = 0.25$ to the deployed

(a) Continuous, monotone feedback

(b) Monotone feedback with jump

(c) Continuous, non-monotone feedback

(d) Non-monotone feedback with jump

(e) No feedback

(f) Jump only

Figure 2: Testing the proposed feedback detection method on some simulation examples. We plot actual predictions in probability space on the $x$-axis against feedback in log-odds space on the $y$-axis. The dashed black line is the true feedback; the solid red line is our feedback estimate along with point-wise error bars stretching 1 SE in each direction. Note that in panel (e) the $y$-axis has a much finer scale than in the other panels.

predictions. In order to mimic real datasets, we made our simulation highly imbalanced: There were many strong predictions for the negative class with $\mu_i \ll 0$, but less so for the positive class. This is why our model performed better near $x = 0$ than near $x = 1$.

We fit both the trend $\mu(\cdot)$ and the feedback function $f(\cdot)$ as the sum of a natural spline with $df = 3$ degrees of freedom and knots spread evenly over $[-3, 3]$, and a jump at zero log-odds (i.e., $x = 0.5$). The dashed lines show the different feedback functions $f$ used in each example.

As emphasized earlier, the intercept of the feedback function $f$ is not identifiable from our experiments. We fixed the intercept by setting the average fitted feedback over all training examples to 0. Since all our training sets were heavily imbalanced, this effectively amounted to setting feedback to 0 at $x = 0$. The plots that do not hit the $(0, 0)$ point are missing a sharp spike at the left-most end; the plot ends at $x = \text{logit}(-3) \approx 0.05$.

As we see from Figure 2, our method accurately fits the feedback function in all six examples, including the null case with no feedback. The error bars depict standard asymptotic error bars produced by the R function `lm` when fitting (16).

## C  Extensions and Further Work

In this section, we discuss some possible extensions to the work presented in this paper.

### C.1  Feedback Removal

If we detect feedback in a real-world system, we can try to identify the root causes of the feedback and fix the problem by removing the feedback loop. That being said, a natural follow-up question to our research is whether we can automatically remove feedback. In the context of the linear feedback model (2), we incur an expected squared-error loss of

$$\tilde{\ell} = \beta^2 \, \mathbb{E}\left[\left(\hat{y}_i^{(t)}\right)^2\right]$$

from completely ignoring the feedback problem. Meanwhile, if we use the maximum likelihood estimate $\hat{\beta}$ to correct feedback, we suffer a loss

$$\ell_{\sigma_\nu} = \text{Var}\left[\hat{\beta}\right] \mathbb{E}\left[\left(\hat{y}_i^{(t)}\right)^2\right] + \sigma_\nu^2,$$

where the first term comes from our errors in estimating $\hat{\beta}$ and the second comes from the extra noise we needed to inject into the system in order to detect the feedback.

An interesting topic for further research would be to find how to optimally set the scale $\sigma_\nu$ of the artificial noise under various utility assumptions, and to understand the potential failure modes of feedback removal under model misspecification. In order to remove feedback, we would also need to have some way of dealing with the intercept term.

### C.2  Covariate-Dependent Feedback

Our analysis was presented in the context of the additive feedback model

$$\hat{y}_i^{(t+1)}[\check{y}_i^{(t)}] = \hat{y}_i^{(t+1)}[\varnothing] + f\left(\check{y}_i^{(t)}\right).$$

In practice, however, we may want to let feedback depend on some other covariates $z$

$$\hat{y}_i^{(t+1)}[\check{y}_i^{(t)}] = \hat{y}_i^{(t+1)}[\varnothing] + f\left(\check{y}_i^{(t)}, z_i^{(t)}\right);$$

for example, we may want to slice feedback by geographic region. One particularly interesting but challenging extension would be to make feedback depend on the unperturbed prediction $\hat{y}_i^{(t)}[\varnothing]$:

$$\hat{y}_i^{(t+1)}[\tilde{y}_i^{(t)}] = \hat{y}_i^{(t+1)}[\varnothing] + f\left(\tilde{y}_i^{(t)},\, \hat{y}_i^{(t)}[\varnothing]\right).$$

For example, if $\hat{y}$ is a prediction for how good a search result is, we might assume that search results that are actually good $(\hat{y}[\varnothing] \gg 0)$ have a different feedback response from those that are terrible $(\hat{y}[\varnothing] \ll 0)$. The challenge here is that $\hat{y}[\varnothing]$ is unobserved, and so we need to have it act on $f$ via proxies. Developing a formalism that lets $f$ depend on $\hat{y}[\varnothing]$ in a useful way while allowing for consistent estimation seems like a promising pathway for further work.

### C.3 Penalized Regression

The key technical challenge in implementing our method for feedback detection is solving the spline equation (12). In Section 4 we proposed a pragmatic approach that enabled us to get good feedback estimates in many examples. However, it should be possible to devise more general methods for fitting $f$. The equation (12) is linear in $f$, and so any strictly convex penalty function $L : \mathcal{A} \to \mathbb{R}$ over some convex subset $\mathcal{A} \subseteq \{\mathbb{R} \to \mathbb{R}\}$ of real valued functions on $\mathbb{R}$ leads to a well-defined estimator $\hat{f}$ through the convex optimization problem

$$\hat{f}_L = \mathrm{argmin}_{f \in \mathcal{A}} \left\{ \sum \left( \hat{y}_i^{(t+1)}[\hat{y}_i^{(t)} + \nu_i^{(t)}] - \mu\left(\hat{y}_i^{(t)}\right) - f\left(\hat{y}_i^{(t)} + \nu_i^{(t)}\right) \right. \right. \tag{22}$$

$$\left. \left. + \varphi_N * f\left(\hat{y}_i^{(t)}\right) \right)^2 + L(f) \right\}.$$

In the context of smoothing splines, a popular choice is to use

$$L(f) = \lambda \int_{\mathbb{R}} \|f''(x)\|^2 \; dx$$

and make $\mathcal{A}$ be the set on which this integral is well-defined. There is an extensive literature on non-parametric regression problems constrained by smoothness penalties [16, 17, 18, 19, 20]; presumably, similar approaches should also give us smoothing spline solutions to (12).

### C.4 Deterministic Designs

Finally, in this paper, we have focused on detecting feedback by adding random noise to raw model predictions. It would be interesting to see whether we can improve the efficiency of our procedure by optimizing the noise choice more closely and using a deterministic design. The problem of finding optimal designs for spline-type problems has been studied by several authors [21, 22, 23].

## D  Proofs

*Proof of Theorem 1.* Because $\nu_i^{(t)}$ is fully artificial noise, we know a-priori that $\nu_i^{(t)}$ and $\hat{y}_i^{(t+1)}[\hat{y}_i^{(t)}]$ are independent. Thus, we can treat $\hat{y}_i^{(t+1)}[\hat{y}_i^{(t)}]$ as a homoscedastic noise term for our regression, and (3) follows immediately from standard results for ordinary least squares regression.  $\square$

*Proof of Theorem 2.* The $\eta_i^{(t)}$ are independent of the $\nu_i^{(t)}$, and so (7) follows from an argument analogous to the one that led to (3). If the $\eta_i^{(t)}$ are still homoscedastic after conditioning on $\hat{y}_i^{(t)}$ then, because the $\eta_i^{(t)}$ are mean-zero by construction, the fact that $\hat{\beta}^*$ is the best linear unbiased estimator of $\beta$ follows directly from an application of the Gauss-Markov theorem where we treat $\nu_i^{(t)}$ as fixed and $\eta_i^{(t)}$ as random, see[24], p. 184.  $\square$

*Proof of Theorem 3.* Given the regression problem described above, (13) follows directly from standard results on heteroscedastic linear regression [9, 10]. Note that our theoretical result assumes that $\hat{\beta}_\mu$ and $\hat{\beta}_f$ are trained on independent data sets. □