[Reviews · NeurIPS 2014]

Submitted by Assigned_Reviewer_8

This paper explores the problem of feedback detection for live/online predictors. In the problem setting, an online predictor makes predictions at every time step by taking historical data for training. However, the historical data is not i.i.d. and could be influenced by the previous predictions the predictor made. Therefore, the predictor could be biased by itself over time. In this work, the main goal is to detect whether this prediction bias, i.e., “feedback”, exists in the system. To solve this problem, the authors propose to add small random noise in the prediction at every time step. Given the additive-feedback assumption, the authors show that the feedback can be detected by running regression over the added noise at time t and the predictions at time t+1 for all t. The authors also run simulations by adding feedback rules in the real-world data.

I like that the paper is trying to solve a real-world problem. The idea of adding noise to the predictions is also interesting.

However, due to the strong assumption the authors make, both the techniques and the results are not too interesting nor novel. In particular, the authors assume the feedback is additive, this makes the analysis much simpler by essentially running regression to fit the feedback function. While the authors did discuss about different forms of feedback functions, the technique is similar.

I am also wondering why standard exploration-exploitation approaches for online learning are not applied here. I understand that, in real systems, we would want the predictions to be as good as possible at every time step. However, if the feedback does exist, the predictor could just go wrong anyway. It makes more sense to apply online learning approaches in the beginning.

Summary: To sum up, although the paper aims to solve a real problem, the techniques and results are not novel nor interesting. I would lean towards rejecting this paper.

Submitted by Assigned_Reviewer_10

** I have read the author feedback. Any paragraphs preceded by (**) were added after taking into account this feedback. My overall view of the paper has not changed; I think the problem is very interesting and the paper makes a novel conceptual contribution with a robust, practical algorithm. The theoretical contribution is limited and the experimental performance examination could be more thorough - these issues prevent me from assigning the paper a higher score **.

Summary:

The paper examines the problem of determining whether there is a feedback loop between historical predictions made by a predictor and current predictions. The authors propose the introduction of randomization in the predictor output to allow the detection of non-linear feedback. The paper reports results of a pilot study derived from a predictive system that is currently used in a search engine.

Strengths:

(1) The paper addresses a very interesting problem that has important practical implications and has not received an enormous amount of attention in the past literature. The technique that is proposed is novel and effective.

(2) The paper is well-written. The method is clearly explained and there is consideration of both practical implementation and theoretical characterization.

Weaknesses:

(1) The theoretical results are all straightforward. Proofs that amount to 3-4 lines of argument followed by statements along the lines of “follows immediately from standard results” mean that the results really don’t warrant the “Theorem” label. I think the results would have been more interesting if they addressed the case where prediction was made from multiple time-steps, which seems to be a setting more closely related to the practical setting. The stated relationships are useful in providing an indication of what factors of the problem will dictate performance, and I am not belittling their value with this comment, but the theoretical contribution is relatively modest.

** In their response, the authors acknowledge that the contribution is primarily conceptual, rather than theoretical. They state that there was a deliberate attempt to “run everything using linear regression” to increase robustness and credibility in an industrial setting. This is a reasonable approach. I agree that the conceptual contribution and novelty are strengths of the paper; in addition to this, one would hope for either a significant theoretical contribution or a thorough experimental investigation. In my opinion, the paper lacks both of these, so I view it as lying above the acceptance threshold, but only marginally so.

(2) Although it is nice to see a discussion of a pilot study with a real system, much of the information provided about the study is vague. While I can imagine that some of the information might be commercially sensitive, with such little information it’s hard to assess the value of the results. For example, the injected feedback is only described as “half a dozen rules of the form…” and although the variance of the injected noise is provided, we aren’t informed of the variance of the natural noise, so it is difficult to assess whether this represents a high level of injected noise or a very small distortion. There is no assessment of how the performance of the technique varies in this setting as any parameter is varied (noise level, number of data points, number of time steps, etc.) so the study offers a very limited characterization of performance. I realize that further simulations are provided in the supplementary material, but the main paper contains virtually no discussion of the observations derived from those simulations (which also do not really explore the impact of various algorithmic parameters).

** The authors explain that the company refused to allow more expansive disclosure of the pilot study and the operation of the classifier. This is reasonable, but I think it strongly motivates the inclusion of a simulation study in the main paper that provides a more thorough characterization of the proposed method.
Summary: The paper a very interesting problem that has important practical implications and has not received an enormous amount of attention in the past. The technique that is proposed is novel and effective. The theoretical characterization is modest in nature and the provided pilot study provides limited indication of the capabilities and sensitivities of the method.

Submitted by Assigned_Reviewer_43

Summary:
The paper proposes a method for detecting feedback loops. Feedback loops occur is system where predictions now effect future predictions. The algorithm can detect both linear and non-linear feedback dependence. For linear feedback it detects it by adding noise to the previous predictions and then learns the linear dependence by regressing on the next prediction using linear least squares. For non-linear feedback the problem of estimating the feedback reduces again to a non-parametric least squares problem.

Detailed feedback:
The authors should elaborate more on the proof of theorem 1 in the appendix.

How realistic is the solution in theorem 1. Could you give examples where the assumption that multiple predictions happen at the same time holds?

The efficiency and conditioning results are not intuitively obvious. Not sure how Equation 8 is derived.

Overall, derivation and solution of the basic formulas 8 and 16 are dense and hard to follow.

Does the algorithm handle feedback detection over multiple steps? Does the algorithm and experiments support this? Not sure this is true in the experiments.
Summary: The paper is original and describes an interesting problem and solution. It is somewhat dense though and hard to appreciate the derivation of the two basic formulas and solutions. The experimental section also needs more clarity.

Submitted by Assigned_Reviewer_44

The authors study situations in which predictions influence future inputs of
the predictor, thereby introducing feedback into the system. The authors assume
that the predictor is linear in terms of its inputs, and model the feedback as
a term that is added to the new prediction at $t+1$ and may depend nonlinearly
on the old prediction at $t$. The authors show that under these assumptions, it
is possible to estimate the feedback function by adding independent noise to
the prediction. The method is similar in spirit to the "randomization" used in
randomized clinical trials, or in other situations in which the goal is to
identify a causal effect (instead of merely estimating statistical
association). The independence of the added prediction noise allows one to
estimate the desired causal effect (which is the feedback, in this case).

Although it is always difficult with articles like these to judge the novelty
(as similar ideas might have come up in very different disciplines, ranging
from econometry, engineering, control theory to causality as a subdiscipline of
AI and/or statistics), I haven't seen similar ideas before in the context of
learning systems.

The most relevant related work that addresses feedback loops is
all the work about "position bias" in the learning-to-rank literature. A common way to address feedback loops is to use interleaved comparison methods. This exploits the fact
that if the prediction is a ranking, one can compare two different rankers by only outputting a single ranking (which combines, in an interleaved way, the rankings obtained by the two rankers that are compared), thereby allowing to partially observe factual and counterfactuals simultaneously.

Nevertheless, the problem setting is interesting and the way in which it is formalized seems to be original. The analysis by the authors seems technically correct (although I didn't check the Theorems in detail) and I enjoyed reading the article. As I think the problem of feedback in live predictor systems is an interesting one that may become
more relevant in the future (as machine learning gets more widely deployed) and
may deserve more attention in the NIPS community, I recommend acceptance.

Two aspects with respect to which the article can still be improved are:
(1) convincing the reader more of the significance of the problem that is addressed;
(2) being more explicit about the assumptions that are made. For example, it would be very instructive to show how one could think about the additive-feedback assumption as a first
order approximation (by some Taylor expansion argument). Also, what assumptions are made about the "natural process" $x^{(t)}$? It seems to me that the authors assume some kind of Markov property. In lines 211-214, the authors make the assumption that the predictor is linear. Is that assumption really necessary, or is the only relevant starting point for their analysis the equation in line 212-213 (basically, the additive feedback equation), which could possibly be interpreted as a first-order approximation of some more complicated nonlinear system (thereby allowing for nonlinear predictors)?

Some small comments:

o Why call this "statistical feedback"? Wouldn't something like "predictor
feedback" be a more descriptive term?

o Lines 44-45: "This paper proposes a fully automatic method for detecting
statistical feedback loops that does not require modeling the feedback
mechanism." This sentence is misleading, as the authors DO model the feedback
mechanism (as an additive term that may depend nonlinearly on the last
prediction, see line 211). Another assumption about the feedback process that
the authors make is that the prediction $\hat y^t$ only directly influences
$x^{t+1}$ (it could also directly influence $x^{t+i}$ for any $i > 1$). I
therefore suggest reformulating or weakening this claim.

o Line 47: "If future model predictions increase when we add artificial noise
to the current ones, then our system has a feedback problem." This sentence is
confusing. It would help to replace the word "increase" by "change" (because
the sign of the change depends on the noise value - even assuming a positive
correlation between noise and feedback, then the future model predictions only
increase when the noise is positive, and they decrease when the noise is
negative). Also, it is not clear why the feedback is a "problem". As shown in
the pilot study, feedback is not necessarily a problem (although it may become
one if the feedback is too strong).

o Line 211: what do you mean by "feedback function in $x$-space"?

o Line 213: shouldn't this be $f(\cdot) = w \cdot f_{(x)}(\cdot)$?

o (10): for consistency, use $\phi_N$ instead of $\phi_{\sigma_\nu}$
Summary: Interesting and original problem setting, decent first attempt at addressing it. If the authors would have convinced me more that the problem they study is significant, I would have given a higher score (now it's up to the reader to guess how significant this contribution is).
Author Feedback
Author rebuttal: Thank you for your constructive feedback!

--------

R1: Thank you for your helpful comments. We agree - it is surprising how little prior work exists on this kind of feedback. The problem discussed in this paper had been bothering the search ads team at our company for quite a while. Concretely, the search ads team has a classifier that will tell you whether a search-query/ad pair has a certain property (call it shoppiness). Now, other teams in the company (such as maps, shopping, etc.) are also interested in knowing whether some query/ad pairs are shoppy, and would want to modify how they display the ads accordingly. However, search ads is concerned that these other teams using the shoppiness classifier might create feedback loops that break the classifier; this concern has in fact delayed some collaborations with other teams.

The present paper came out of a research collaboration aiming to make some headway on this problem. The main breakthrough was conceptual, not technical. The feedback model (1) may appear very simple, but before arriving at it we spent months trying other definitions of feedback that lead to completely intractable estimation problems. Once we have the definition (1) and the idea to add artificial noise, the technical steps needed to get to the estimation algorithm (16) are straight-forward (given decent notation).

Concerning the examples - the company was adamant in not letting us describe the real-world pilot study in more detail (as SEO spammers who gained insight about the classifier could steal lots of money). We were hoping that the simulation study in the appendix would be helpful by having more details. In our revision, we plan to add a detailed example described in the response to R3.

--------

R2: Thank you for your encouraging feedback! The assumptions made in Theorem 3 (of which Theorem 1 is a special case) are fairly realistic for our real-world application. Suppose, for example, that our predictions depend on click-through-rate (CTR) averaged over a 1-week period and that these numbers are updated as a batch each week. Then all the ad/query pairs experience a simultaneous change in CTR each week. Note that our method also works if different predictions don't occur at the same time; however, the notation would get even more verbose.

We will make an effort to provide more details about the math in our revision. Our algorithm is actually agnostic as to whether predictions occur over multiple time steps. All we need for our algorithm is tuples of the form (previous prediction, previous noise, current prediction). The timestamp of the tuples does not matter.

--------

R3: Thank you for your constructive remarks. As discussed in our response to R1, we agree that the technical devices used in our paper are standard: we made an effort to run everything using linear regression. Our contribution is not technical but rather conceptual: although our feedback model is simple, it took a long time to get it just right (and we also first tried many other approaches that lead to inferential nightmares).

The fact that the techniques used by our paper are simple was in fact critical for the algorithm to be usable in practice. The team managers (who have PhDs in statistics or related fields) are much more conservative than academics in the algorithms they are willing to consider. Having new and untested statistical algorithms run wild on live servers could lead to huge financial losses. Thus, that we were able to frame our approach in terms of linear regression helped people believe that our method would in fact reliably work.

Finally, studying feedback in an online exploitation exploration framework would be interesting. The current problem we are working on, however, does not allow for online learning, as explained below. We plan to add an example following the one below to the text.

Our problem is as follows:

- At time 0, we collect, say, 100,000 ad/query pairs, and ask a panel of human raters to score them as not shoppy or shoppy.

- Based on these 100,000 examples, we train a logistic regression classifier for shoppiness. Suppose we use CTR as a feature.

- Now, we have a shoppiness classifier running in the background, and suppose it works well. Thus, we now know which ad/query pairs are shoppy or not.

- Then, at time t0, an engineer hears about the classifier, and wants to use the shoppiness classifier for display purposes (e.g., by adding a seller rating to shoppy ads).

- Suppose we let the engineer do so, and it turns out that adding seller ratings increases CTR. Now, ads that get classified as shoppy get a boosted CTR, which makes them look even shoppier, etc. This leads to 2 problems: First, our background shoppiness classifier (which was very important for business analytics) is broken; second, borderline shoppy results may drift away from the boundary in unpredictable ways.

- Our method would let us detect this problem. Once we had detected this problem, we could work with the engineer to make it go away. However, if we don't have a detector in place, we could have rampant feedback without anyone knowing.

- Finally, we could not address this problem by online learning. This is because we only have labeled data at t = 0; after that, we can only observe the algorithm run but do not know whether or not it is making correct predictions. Thus, we don't have any responses for online learning to use.

Pulling everything together - the key problem is that, once the classifier is live, we really don't have a good systematic way of seeing how well it is doing. This is why we cannot do online learning, and this is why having automatic feedback detectors is so important.